# Diagnostic Accuracy of Bone Scintigraphy for the Histopathological Diagnosis of Cardiac Transthyretin Amyloidosis—A Retrospective Austrian Multicenter Study

**DOI:** 10.3390/biomedicines10123052

**Published:** 2022-11-28

**Authors:** Nicolas Verheyen, Maria Ungericht, Lisa Paar, Kathrin Danninger, Stefanie Schneiderbauer-Porod, Franz Duca, Bernhard Cherouny, Viktoria Hoeller, Klemens Ablasser, David Zach, Ewald Kolesnik, Daniel Kiblboeck, Matthias Frick, Diana Bonderman, Josef Dierneder, Christian Ebner, Thomas Weber, Gerhard Pölzl

**Affiliations:** 1Division of Cardiology, Department of Internal Medicine, Medical University of Graz, A-8036 Graz, Austria; 2Department of Internal Medicine III, Cardiology & Angiology, Medical University of Innsbruck, A-6020 Innsbruck, Austria; 3Division of Gastroenterology and Hepatology, Department of Internal Medicine, Medical University of Graz, A-8036 Graz, Austria; 4Department of Internal Medicine II, Cardiology and Intensive Care, Klinikum Wels-Grieskirchen, A-4600 Wels, Austria; 5Department of Internal Medicine 2, Hospital of the Order of St. Elizabeth, A-4020 Linz, Austria; 6Department of Cardiology, Medical University of Vienna, A-1090 Vienna, Austria; 7Department of Cardiology, Kepler University Hospital, Med Campus III, Medical Faculty, Johannes Kepler University, A-4020 Linz, Austria; 8Department of Internal Medicine I, Teaching Hospital Feldkirch, A-4020 Feldkirch, Austria; 9Division of Cardiology and Emergency Medicine, Favoriten Clinic, A-1100 Vienna, Austria; 10Department of Nuclear Medicine, Hospital of the Order of St. Elizabeth, A-4020 Linz, Austria

**Keywords:** cardiac amyloidosis, bone scintigraphy, transthyretin cardiomyopathy, diagnostic accuracy, endomyocardial biopsy, retrospective cohort study

## Abstract

We aimed to ascertain the real-world diagnostic accuracy of bone scintigraphy in combination with free light chain (FLC) assessment for transthyretin (ATTR) cardiac amyloidosis (CA) using the histopathological diagnosis derived from endomyocardial biopsy (EMB) as a reference standard. We retrospectively analyzed 102 patients (22% women) with suspected CA from seven Austrian amyloidosis referral centers. The inclusion criteria comprised the available results of bone scintigraphy, FLC assessment, and EMB with histopathological analysis. ATTR and AL were diagnosed in 60 and 21 patients (59%, 21%), respectively, and concomitant AL and ATTR was identified in one patient. The specificity and positive predictive value (PPV) of Perugini score ≥ 2 for ATTR CA were 95% and 96%. AL was diagnosed in three out of 31 patients (10%) who had evidence of monoclonal proteins and a Perugini score ≥ 2. When excluding all patients with detectable monoclonal proteins (n = 62) from analyses, the PPV of Perugini score ≥ 2 for ATTR CA was 100% and the NPV of Perugini score < 2 for ATTR CA was 79%. Conclusively, ATTR CA can be diagnosed non-invasively in the case of a Perugini score ≥ 2 and an unremarkable FLC assessment. However, tissue biopsy is mandatory in suspected CA in any other constellation of non-invasive diagnostic work-up.

## 1. Introduction

Cardiac amyloidosis (CA) is an infiltrative storage disease causing progressive heart failure [1]. It is caused by the extracellular deposition of amyloid fibrils in the myocardium and in the majority of cases occurs due to transthyretin (ATTR) or immunoglobulin light chain (AL) amyloidosis. Without modern therapies, CA is associated with a 5-year mortality rate of almost 60% and is considered the most malignant cardiomyopathy, with AL carrying an even worse prognosis than ATTR [2]. With the recent introduction of specific amyloid-directed therapies both for AL and ATTR, affected patients can be provided with efficient treatment regimens improving symptoms and prognosis [3,4]. Late diagnosis is associated with poor treatment outcomes [5].

Although the diagnosis of AL requires organ biopsy with the typing of deposits, it can be ruled out non-invasively using free light chain (FLC) assessment [6]. ATTR CA, however, can be confirmed non-invasively. Previous studies indicated that the cardiac uptake of bone-avid technetium-labeled tracer during scintigraphy with a Perugini score ≥ 2 yields a positive predictive value (PPV) of 100% for ATTR CA when AL CA has been ruled out using FLC assessment [7]. Consequently, guidelines prioritize a non-invasive diagnostic approach over the actual gold-standard diagnostic method of endomyocardial biopsy (EMB) to diagnose ATTR CA [6,8]. Whether this degradation of EMB leads to misdiagnoses in a real-world setting is currently under-investigated. We aimed to ascertain the real-world diagnostic accuracy of bone scintigraphy in combination with FLC assessment for ATTR CA in patients with suspected cardiac amyloidosis using the histopathological diagnosis derived from EMB as a reference standard.

## 2. Materials and Methods

This was a multicenter retrospective analysis including patients with suspected CA and diagnostic work-up conducted between 2017 and 2020 in Austria. The inclusion criteria comprised available results of (1) bone scintigraphy, (2) FLC assessment, and (3) EMB with histopathological tissue analysis. Ten contacted referral centers for cardiac amyloidosis throughout Austria performed a systematic retrospective medical record search and seven sites provided data from 109 patients, of whom 102 fulfilled all inclusion criteria. Scintigraphy was performed using a 99 m technetium-labeled tracer and planar acquisition. Cardiac tracer uptake was visually graded using the Perugini score, into grade 0 (no uptake), grade 1 (faint uptake), grade 2 (uptake similar to ribs), and grade 3 (uptake more than ribs) [9]. EMB was performed either from right or left ventricular access, depending on the local standard. Myocardial samples were fixed in formaldehyde, embedded in paraffin, and tested for the presence of amyloid via Congo-red staining and apple-green birefringence under cross-polarized light microscopy. Amyloid subtyping was achieved by immunohistochemistry or mass spectrometry, respectively. The study was approved by local ethics committees and adheres to the principles of Good Clinical Practice and the Declaration of Helsinki.

FLC assessment included the quantification of the serum kappa-to-lambda ratio and immunofixation electrophoresis (IFE) in both serum and urine. The estimated glomerular filtration rate (eGFR) was calculated using the Modification of Diet in Renal Disease formula. The presence of monoclonal protein was defined by an abnormal FLC ratio (<0.26 or >1.65 for eGFR > 55 mL/min, <0.82 or >3.6 for eGFR ≤ 55 mL/min) or documented monoclonal band in serum or urine IFE [10]. 

Categorical variables are expressed as counts with percentages, continuous variables as means ± standard deviation or medians with quartiles, as appropriate. Diagnostic accuracy read-outs of Perugini score for EMB-derived amyloid subtyping were expressed as sensitivity, negative predictive value (NPV), specificity, and PPV. A *p*-value < 0.05 was considered statistically significant. All statistical analyses were performed using IBM SPSS Statistics Version 27 (IBM Corporation, Armonk, NY, USA).

## 3. Results

### 3.1. Baseline Characteristics

The study cohort consisted of 102 patients (22% women) and the number of patients provided per center ranged from 2 to 46. The number of patients per year increased from 15 patients in 2017 to 32 patients in 2020. Median age was 75 (68–78) years and median N-terminal B-type natriuretic peptide (NT-proBNP) was 2652 (1580–5167) pg/mL. Among 85 patients with Congo-red-positive EMBs (83%), ATTR CA was diagnosed in 60 patients (71%) and AL CA in 21 patients (25%), and one further patient had concomitant AL and ATTR. Other diagnoses included ApoA4 amyloidosis (n = 2) and AA amyloidosis (n = 1), whereas amyloidosis was ruled out in 17 patients (20%). Among patients with ATTR, 2 (3%) had the hereditary form (hATTR) and 59 (97%) had the wild-type form (wtATTR). One patient with hATTR had two variants in *TTR* (NM_000371:c.76G>A, p.G26S and NM_000371:c.114T>G,p.D38E) and another patient had one mutation (NM_000371:c.206C>T, p.T69I).

### 3.2. Bone Scintigraphy and FLC Assessment

Scintigraphy revealed a Perugini score of 0/1 in 45 patients (44%) and a Perugini score of 2/3 in 57 patients (56%). Single-photon emission computed tomography (SPECT) was performed in 56% of scintigraphies. FLC assessment was indicative of monoclonal protein in 62 patients (61%) and unremarkable in 40 patients (39%). Among these 62 patients with FLC assessment indicative of monoclonal protein, 11 (18%) had monoclonal bands, 14 (23%) had an abnormal FLC ratio, and 37 (60%) had both a monoclonal band and an abnormal FLC ratio. Of those with monoclonal bands (n = 48), there were 22 (46%) with an isolated monoclonal band in their serum, 2 (4%) with an isolated monoclonal band in their urine, and 24 (50%) with monoclonal bands both in their serum and urine. Patients with a Perugini score ≥2 compared to those with a Perugini score < 2 had evidence of monoclonal bands less often (33% vs. 64%, *p* = 0.003), were older (median age 77 (73–80) vs. 70 (60–76) years, *p* < 0.001), and were less often women (11% vs. 36%, *p* = 0.002). Among the two patients with hATTR, the patient carrying two mutations had evidence of monoclonal protein due to an abnormal FLC ratio of 1.757 (eGFR of 78 mL/min/1.73 m²), whereas the other mutation-carrier had unremarkable FLC assessment results. Further baseline characteristics are shown in Table 1.

### 3.3. Diagnostic Accuracy Analyses

Among 61 patients with ATTR CA, 55 had a Perugini score ≥ 2, yielding a sensitivity of Perugini score ≥ 2 for ATTR of 90%. Among the two patients with hATTR, the patient carrying two mutations had a Perugini score of 3, and the other mutation-carrier had a Perugini score of 2. Among 41 patients without ATTR, Perugini score was <2 in 39 of the patients. A Perugini score ≥ 2 were found in 57 patients, of whom 55 had ATTR (including one patient with concomitant AL and ATTR). Thus, the specificity and PPV of Perugini score ≥ 2 for ATTR CA were 95% and 96%. Counts of EMB-derived diagnoses stratified by Perugini score are illustrated in Figure 1, and the according diagnostic accuracy read-outs are shown in Table 2. AL was diagnosed in 21 patients with evidence of monoclonal protein (including one patient with concomitant AL and ATTR) compared with one patient with AL among those with normal FLC assessment, yielding a sensitivity and NPV of FLC assessment for AL of 95% and 98%, respectively. AL was diagnosed in three of 31 patients (10%) with FLC assessment indicative of monoclonal protein and a Perugini score ≥ 2 (including one patient with concomitant AL and ATTR). When all patients with detectable monoclonal protein were excluded from analysis (n = 62) and only those with unremarkable FLC assessment results (n = 40) were analyzed, 26 patients with ATTR CA had a Perugini score ≥ 2 and 3 patients with ATTR CA had a score < 2. Counts of EMB-derived diagnoses stratified by Perugini score in patients with unremarkable FLC assessment results are illustrated in Figure 2, and the according diagnostic accuracy read-outs are shown in Table 3. The PPV of Perugini score ≥ 2 for ATTR CA was 100%, whereas the NPV of Perugini score < 2 to exclude ATTR was 79% (Table 3).

## 4. Discussion

In a retrospective, multicenter study including patients with suspected cardiac amyloidosis, bone scintigraphy with Perugini score ≥ 2 yielded 100% specificity and 100% PPV for cardiac ATTR amyloidosis if AL was ruled out by means of FLC assessment. However, 10% of patients with ATTR and unremarkable FLC assessment results had a Perugini score < 2, and a Perugini score of <2 had an NPV of 79% for ATTR CA. These data reflect clinical practice throughout Austria and as such provide contemporary and novel insights into the diagnostic accuracy of bone scintigraphy when adopted in a real-world setting. The data confirm that ATTR CA can occur with all grades of myocardial tracer uptake during scintigraphy. Particularly at early stages with a less severe extent of amyloid load, ATTR CA can be associated with no or faint myocardial tracer uptake [11]. Of note, the sensitivity of Perugini score ≥ 2 was particularly low in patients with CA due to hATTR and the Phe64Leu mutation [12]. In our study, however, this mutation was not present and the overall hATTR proportion among patients with ATTR CA was as low as 3%. It is likely that detection rates of ATTR CA at early and less symptomatic stages will further rise due to increasing awareness of the disease and the better availability of bone scintigraphy [13]. The present real-world data confirm that (1) a Perugini score < 2 is not suited to ruling out ATTR and warrants a tissue biopsy in the case of suspected CA; (2) patients with a Perugini score ≥ 2 and FLC assessment results that are not indicative of monoclonal protein can be diagnosed with ATTR CA without further tissue biopsy.

Confirming the overall excellent diagnostic accuracy of bone scintigraphy for the diagnosis of ATTR CA, the results of our study are in line with existing literature. For instance, in a recent meta-analysis involving 39 studies with a total of 3636 patients, the pooled sensitivity and specificity values of visual scoring using bone scintigraphy for the diagnosis of ATTR CA were 97% and 96%, respectively [14]. Interestingly, among patients with evidence of monoclonal protein, 10% of patients with a Perugini score ≥ 2 had AL amyloidosis as the underlying condition, including one patient with concomitant ATTR and AL. In fact, AL CA can lead to significant myocardial tracer uptake and it is not possible to exclude AL with the use of scintigraphy [15]. In line with this, in another recent study by Nitsche and colleagues, up to 15% of biopsy-proven AL CA had a Perugini score ≥ 2 [16]. One patient in our cohort had AL CA without any evidence of monoclonal protein in the FLC assessment, as reported previously [17]. Our analysis is therefore not fully in line with previous studies reporting 100% sensitivity of FLC assessment indicative of monoclonal protein for the presence of AL, and a 100% NPV of normal FLC to rule out AL [18]. It has also been reported by others that FLC assessment results can be a false negative for AL although underlying mechanisms remain elusive [17,19]. Aside from AL CA, other cardiomyopathies can also lead to faint myocardial tracer uptake, such as chloroquine-induced cardiomyopathy [20,21], myocardial ischemia or edema [22], hypertrophic cardiomyopathy [23], and hereditary apolipoprotein CA [7]. Two patients with ApoA4 CA in our cohort exhibited no myocardial tracer uptake. Our analysis is in line with other previous studies emphasizing that patients with suspected CA and a Perugini score < 2 require particular attention and dedicated diagnostic work-up, including tissue biopsy, in order to facilitate targeted therapy.

The differentiation of CA from other cardiomyopathies with a hypertrophic phenotype can be challenging. Clinically relevant phenocopies include hypertrophic cardiomyopathy, hypertensive heart disease, athlete’s heart, and, rarely, arrhythmogenic right ventricular cardiomyopathy [24,25]. In patients with left ventricular hypertrophy, the careful assessment of symptoms and medical history is mandatory to assess the pre-test probability of CA, before costly diagnostic procedures with limited availability such as cardiac magnetic resonance imaging and bone scintigraphy are initiated. Although carpal tunnel syndrome, lumbar stenosis or biceps tendon rupture are typical features of CA, syncope may point towards hypertrophic cardiomyopathy as an underlying condition [26,27,28]. In experienced hands, speckle-tracking echocardiography can offer additional diagnostic benefit, and typical features (an apical sparing pattern or radial basoapical gradient, and a high EF global longitudinal strain ratio) yielded high diagnostic accuracy for the presence of CA [29,30].

The limitations of our study include its retrospective character, potentially introducing selection bias, which is indicated by the relatively high proportion of patients with evidence of monoclonal protein. This may impair the generalizability of absolute counts reported in our study. The assessment of baseline characteristics such as symptoms, medical history including red flags of CA, and features of echocardiography was beyond the scope of our study, which is another limitation. Scintigraphy tracers varied by site due to different protocols, and scintigraphy was partly based on planar images, because SPECT was not routinely performed at all participating sites during the study period. Nevertheless, the studied population likely covers a vast majority of the patients with suspected CA undergoing both non-invasive diagnostic work-up and EMB in Austria between 2017 and 2020.

## 5. Conclusions

These real-world data acquired throughout Austria confirm that ATTR CA can be diagnosed using bone scintigraphy in the case of unremarkable FLC assessment results. They also underscore that tissue biopsy and histopathological analysis are required in any patient with suspected amyloidosis and evidence of monoclonal protein, irrespective of the scintigraphic result. Given that a Perugini score < 2 is not suited to rule out ATTR CA, tissue biopsy should be considered in any patient with suspected CA in the case of a Perugini score < 2.

## Figures and Tables

**Figure 1 biomedicines-10-03052-f001:**
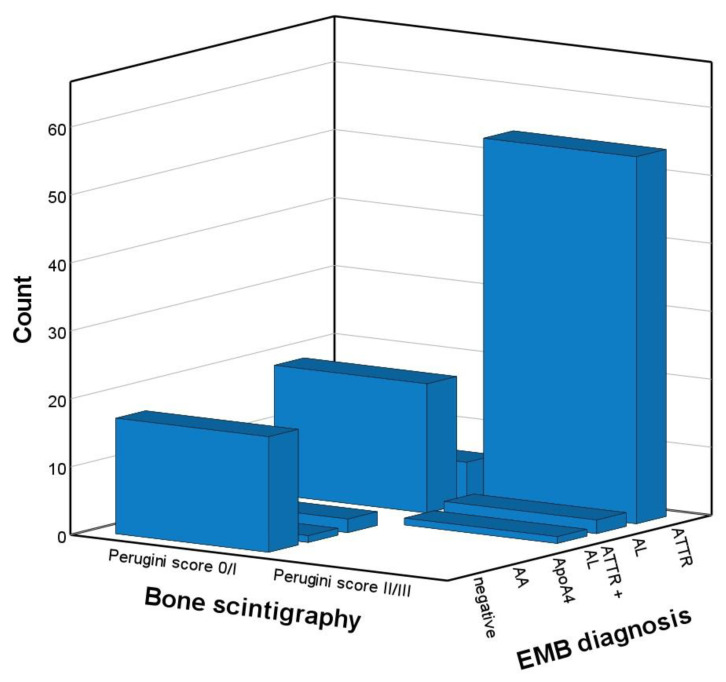
Endomyocardial biopsy-derived diagnoses stratified by Perugini score of 0/1 vs. 2/3 in the total sample (n = 102). AA, amyloid A; ApoA4, apolipoprotein A4; ATTR, transthyretin amyloidosis; AL, light chain amyloidosis; EMB, endomyocardial biopsy.

**Figure 2 biomedicines-10-03052-f002:**
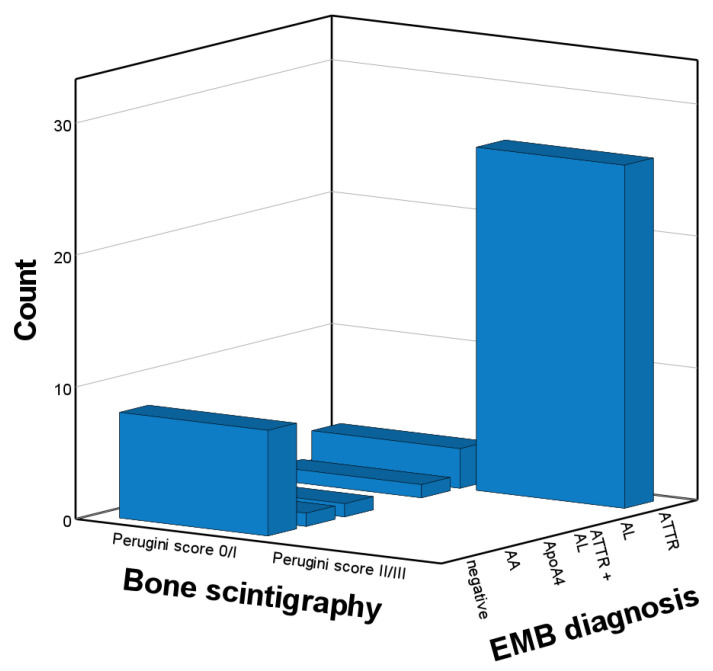
Endomyocardial-biopsy-derived diagnoses stratified by Perugini score of 0/1 vs. 2/3 in patients without monoclonal protein (n = 40). AA, amyloid A; AL, light chain amyloidosis; ApoA4, apolipoprotein A4; ATTR, transthyretin amyloidosis; EMB, endomyocardial biopsy.

**Table 1 biomedicines-10-03052-t001:** Baseline characteristics of the total sample and stratified by Perugini score.

	All (N = 102)	Perugini Score	
		0/1 (N = 45)	2/3 (N = 57)
Age, years	75 (68–78)	70 (60–76)	77 (73–80) **
Women, n (%)	22 (22%)	16 (36%)	6 (11%) *
NT-proBNP, pg/mL	2652 (1580–5167)	2966 (1322–5205)	2578 (1624–5183)
eGFR MDRD, mL/min/1.73 m²	60 ± 19	58 ± 18	61 ± 19
**Bone scintigraphy**			
SPECT	57 (56%)	29 (64%)	28 (49%)
Tracer, n (%)			
DPD	87 (85%)	35 (78%)	52 (91%)
HDP	6 (6%)	3 (7%)	3 (5%)
PYP	9 (9%)	7 (16%)	2 (4%)
Perugini score, n (%)			
0	31 (30%)	31 (69%)	0 **
1	14 (14%)	14 (31%)	0
2	16 (16%)	0	16 (28%)
3	41 (40%)	0	41 (72%)
**FLC assessment**			
Monoclonal protein, n (%)	62 (59%)	31 (69%)	31 (54%)
Monoclonal band	48 (47%)	29 (64%)	19 (33%) *
serum	46 (45%)	28 (64%)	18 (33%) *
urine	26 (26%)	14 (33%)	12 (23%)
Abnormal FLC ratio	51 (50%)	26 (58%)	25 (44%)
Serum FLC ratio	1.50 (0.78–2.02)	1.36 (0.185–2.37)	1.64 (1.00–1.98)
**EMB-derived diagnosis, n (%)**			
ATTR#	60 (59%)	6 (13%)	54 (95%)
AL	21 (21%)	19 (42%)	2 (4%)
ATTR + AL	1 (1%)	0	1 (2%)
ApoA4	2 (2%)	2 (4%)	0
AA	1 (1%)	1 (2%)	0
negative	17 (17%)	17 (38%)	0

Data are frequency (%), mean (SD), or median (quartiles). # Two patients with hereditary ATTR amyloidosis. * *p* < 0.05; ** *p* < 0.001, for group comparison between Perugini score of 2/3 and 0/1. AA, amyloid A; AL, amyloid light chain; ApoA4, apolipoprotein A4; ATTR, amyloid transthyretin; DPD, diphosphono-1,2-propanodicarboxylicacid; eGFR, estimated glomerular filtration rate; EMB, endomyocardial biopsy; FLC, free light chain; HDP, hydroxydiphosphonate; MDRD, Modification of Diet in Renal Disease formula; NT-proBNP, N-terminal prohormone of brain natriuretic peptide; PYP, pyrophosphate; SPECT, single-photon emission computed tomography.

**Table 2 biomedicines-10-03052-t002:** Diagnostic accuracy of bone scintigraphy with Perugini scorse ≥2 for the histopathological diagnosis of cardiac ATTR in the total sample (n = 102).

		Perugini Score				
		0 N = 31	1 N = 14	2 N = 16	3 N = 41	
**EMB result**	**ATTR**	1	5	14	40	**Sensitivity 90%** **Specificity 95%** **PPV 96%** **NPV 87%**
	ATTR + AL	0	0	1 *	0	
	AL	14	5	1	1	
	ApoA4	2	0	0	0	
	AA	1	0	0	0	
	Negative	13	4	0	0	

All 3 patients with AL and a Perugini score ≥2 had monoclonal bands in serum. * one patient with concomitant ATTR and AL was considered as correctly positive. AA, amyloid A; ApoA4, apolipoprotein A4; ATTR, transthyretin amyloidosis; AL, light chain amyloidosis; EMB, endomyocardial biopsy; PPV, positive predictive value; NPV, negative predictive value.

**Table 3 biomedicines-10-03052-t003:** Diagnostic accuracy of bone scintigraphy with a Perugini score ≥2 for the histopathological diagnosis of cardiac ATTR in patients with FLC assessment results that were not indicative of monoclonal protein (n = 40).

		Perugini Score				
		0 N = 9	1 N = 5	2 N = 7	3 N = 19	
**EMB result**	ATTR	0	3	7	19	**Sensitivity 90%** **Specificity 100%** **PPV 100%** **NPV 79%**
	AL	0	1	0	0	
	ApoA4	1	0	0	0	
	AA	1	0	0	0	
	Negative	7	1	0	0	

AA, amyloid A; AL, light chain amyloidosis; ApoA4, apolipoprotein A4; ATTR, transthyretin amyloidosis; EMB, endomyocardial biopsy; FLC, free light chain; NPV, negative predictive value; PPV, positive predictive value.

## Data Availability

The data presented in this study are available on request from the corresponding author.

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
