# Peer review of "Diagnostic Accuracy of Bone Scintigraphy for the Histopathological Diagnosis of Cardiac Transthyretin Amyloidosis—A Retrospective Austrian Multicenter Study"

_biomedicines, 2022, doi:10.3390/biomedicines10123052_

Round 1
Reviewer 1 Report
Interesting, current subject, with practical implications, from real life as specified in the abstract.
The paper provides a basis for a practical approach to patients with cardiac amyloidosis especially for the practicing physician.
The work is addressed to a rare pathology, the patients are not very frequent in ”real life” and precisely for this reason the manuscript acquires scientific value as a reference work that could be the basis of future research.
But, the retrospective nature of the study limits the discussion”s section that should be more extensive.
This section should be improved.

Reviewer 2 Report
The clinical finding described in the work is not considered something radically new, but cardiac amyloidosis is and remains a clinically significant problem with diagnostic difficulties. The considered volume is wide enough to make conclusions based on statistical methods. The authors' instructions on study limitation can be evaluated as correct and corresponding to the level of good scientific practice. The work will certainly arouse interest in the environment of industry experts and interested parties, as well as in the perspective it will reach a high level of citation.
The design of Figure 1 and Figure 2 can be discussed, but this format makes the information transparent and understandable enough. A small suggestion: it is understood in the text that the numerical values ​​of the data shown in Figure 1 can be seen in Tab. 2 and the data in Figure 2 can be seen in Tab. 3, but the value of the article would only be gained if some indication of this correspondence appeared in the text.
Reviewer 3 Report
This article entitled Diagnostic accuracy of bone scintigraphy for the histopathological diagnosis of cardiac transthyretin amyloidosis – a retrospective Austrian multicenter study submitted to Biomedicines journal is a retrospective original study. The aim of study is to evaluate the real-world diagnostic accuracy of bone scintigraphy in combination with free light chain (FLC) assessment for transthyretin (ATTR) cardiac amyloidosis (CA) using the histopathological diagnosis derived from endomyocardial biopsy (EMB) as reference standard. The title is consistent with the presented problem and reflects the main message of the study. The abstract represents all the sections of the entire article. Overall, the scientific objective is relevant and interesting. The article is well written and comprehensive. There are no ethical concerns about this study. The research design is appropriate and the methods clearly explained. The interpretation of the results is clearly presented and adequately supported by the evidence adduced. This study confirms that ATTR CA can be diagnosed non-invasively in case of Perugini score ≥2 and negative FLC assessment, however other clinical scenarios should include tissue biopsy for ATTR confirmation. The references are up-to-date and the most important studies have been cited. Despite the clear limitations of this methodology (small sample size, retrospective character) it presents high scientific value and important to the field of amyloidosis research and in my opinion should be accepted for publication in the Biomedicines journal.
Reviewer 4 Report
I would congratulate with authors for this interesting paper. Results show that ATTR CA can be diagnosed using bone scintigraphy in case of unremarkable FLC assessment, and underscore that tissue biopsy and histopathological analysis is required in any patient with suspected amyloidosis and evidence of monoclonal protein, irrespectively of the scintigraphic result. I have only three minor points in order to improve the manuscript
Authors should cite the paper from Ruan et al that aimed to describe the diagnostic value of bone scintigraphy in ATTR-CA by performing a meta-analysis of multiple studies (please cite https://doi.org/10.1007/s40336-021-00471-8). Actually, bone scintigraphy has an excellent diagnostic performance in ATTR-CA. An accurate diagnosis of ATTR-CA can be made based on the semi-quantitative visual score, quantitative ratios of planar imaging, and cardiac bone-tracer uptake values of SPECT images
Since Hypertrophic cardiomyopathy (HCM) is the most common inherited monogenic cardiac disorder, in discussion authors should more focus on differential diagnosis of left ventricular hypertrophy (DOI: 10.1111/j.1540-8175.2012.01680.x) also focusing on the importance of symptoms at baseline (DOI: 10.1016/j.ijcard.2022.03.028) since data on the current patient population are missing. Please cite this concept in limitations and cite 2 fundamental suggested references
Again, since in cardiac amyloidosis the morphological findings and clinical features are shared also with athlete's heart (DOI: 10.1093/ehjci/jeu158 ,and, DOI: 10.1111/jce.14526) authors do not discuss at baseline patient population's characteristics. Please cite this concept in limitations and cite 2 fundamental suggested references
Round 2
Reviewer 4 Report
Congratulations! Manuscript finally definitely improved
Author Response
We thank the reviewer for the positive statement.